# Reshaping Model Output Space Via Deep Kernel Density Estimation Networks

## Abstract

Traditional classification models are typically optimized solely for their specific training task without considering the properties of the underlying probability distribution of their output space. As the use of these models for downstream tasks becomes more prevalent, it becomes advantageous to have a framework that can transform the output space of such models to a more convenient space without sacrificing performance. In this paper, we introduce DeepKDE, a novel method which enables the transformation of arbitrary output spaces to match more desirable distributions, such as Normal and Gaussian Mixture Models. We explore the properties of the new method and test its effectiveness on ResNet-18 and vision transformers trained on CIFAR-10 and Fashion MNIST datasets. We show that DeepKDE models succeed in transforming the output spaces of the original models while outperforming them in terms of accuracy.

## 1 Introduction

Leveraging pre-trained models to enhance performance in downstream tasks is a widely used technique. In this framework, a large model is trained on abundance of data, and the resulting output can then serve as a starting point for a new task such as classification (Plested & Gedeon, 2022), anomaly detection (Chalapathy & Chawla, 2019), clustering (Ren et al., 2022) and more. However, the optimization of the primary model focuses on refining its architecture, hyperparameters, and training process to best suit the primary task. These optimizations are not designed to match any underlying probability distribution thus resulting with an arbitrary and intractable probability distribution. It would be advantageous to reshape the output space of such models to align with a more convenient underlying probability function. This can enable downstream algorithms to benefit from the knowledge of the feature space distribution.

Reshaping a feature space to match a specific distribution is commonly done in various learning tasks. However, the current approach is usually task-specific and not a generic solution to transform the output of any arbitrary model.

In image generation tasks, deep generative models such as Variational Auto Encoders (VAE) (Kingma & Welling, 2014), Normalizing Flows (Rezende & Mohamed, 2015) and Diffusion Models (Ho et al., 2020), link between the input space and a feature space with a known underlying probability density function, so new data points could be easily sampled during inference. In the context of clustering tasks, the process of combining representation learning using deep neural networks (DNN) together with a clustering method is referred to as deep clustering. Some deep clustering methods aim to generate a feature space that follows a multimodal distribution such as Gaussian Mixture Model (GMM). Notable methods in this category include Xie et al. (2016) and Yang et al. (2017) who proposed methods to construct deep clustering using autoencoder networks. Mukherjee et al. (2018) proposed a methodology to train clustering Generative Adversarial Networks and Jiang et al. (2017) used VAE with GMM as a prior. In unsupervised anomaly detection tasks, optimization was done for deep representation learning along with one class objective such as a hyperplane (Chalapathy et al., 2018) or a hypersphere (Ruff et al., 2018).

Looking at the aforementioned methods, we can identify two modes of operations: the first involves a joint optimization of DNN parameters and reshaping the feature space to meet a target distribution (Yang et al., 2017; Jiang et al., 2017; Chalapathy et al., 2018; Ruff et al., 2018). The second involves a two-step procedure starting with optimizing DNN parameters, followed by refining the feature space to best suit the task (Xie et al., 2016). Given the growing size of the state-of-the-art models

and the increasing complexity of the tuning methods, it is more practical to adopt the two-step mode of operation.

In this paper we introduce the DeepKDE a novel deep method. We take state-of-the-art classification models and use DeepKDE networks to reshape their output space to match underlying probability density functions such as Normal distributions or GMMs while also improving their classification performance. The workflow begins with taking a primary tuned model and projecting the input data to a predefined layer within the model that captures meaningful features. The projected data is then fed as an input to a DeepKDE network. The DeepKDE network is trained with a loss function optimized to match a desired probability density function while incorporating constraints to preserve the information contained in the input data (Figure 1). Our work focuses on classification models but can be extended to other tasks.

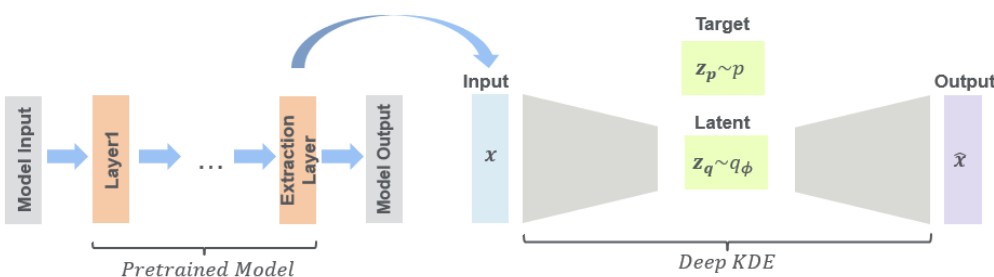

Figure 1: DeepKDE learns a neural network transformation $\Theta\left(\cdot|\phi\right)$ to minimize the statistical distance between $q_\phi$ and the target function $p$, while enforcing the clustering of specific data points within the latent space and encouraging adjacent data points in the input space to remain adjacent in the latent space. The model takes as input the extracted features from a pretrained model.

## 2 DEEPKDE

### 2.1 MODEL BASICS

Consider a typical autoencoder architecture with input and output space $\mathcal{X}, \hat{\mathcal{X}} \subseteq \mathbb{R}^D$ and latent space $\mathcal{Z} \subseteq \mathbb{R}^d$. Let $\Theta\left(\cdot|\phi\right)$ be the autoencoder function with a set of adjustable parameters $\phi$. The latent variable $\mathbf{z}_q \in \mathcal{Z}$ can be interpreted as a random variable with sample space $\mathcal{Z}$ and probability density function $q_\phi$, where $\phi$ denotes the dependency on model parameters. Additionally, suppose we have a source that generates samples $\mathbf{z}_p \sim p$ where $p$ is a known and tractable probability density function on the same sample space $\mathcal{Z}$. Our objective is to optimize the model parameters to minimize the statistical distance between $q_\phi$ and the target function $p$. In this work we measure the statistical distance using the Jensen-Shannon divergence:

$$D_{JS}(q_\phi||p) = \frac{1}{2}\left(\mathbb{E}_{q_\phi}[\log(q_\phi)] - \mathbb{E}_{q_\phi}[\log(m_\phi)]\right)$$
$$+ \frac{1}{2}\left(\mathbb{E}_p[\log(p)] - \mathbb{E}_p[\log(m_\phi)]\right) \quad (1)$$
$$\text{where} \quad m_\phi = \frac{1}{2}(q_\phi + p)$$

By utilizing (1) and considering only the terms dependent on $\phi$, we can define a loss term:

$$L_{pdf} := \mathbb{E}_{q_\phi}[\log(q_\phi)] - \mathbb{E}_{q_\phi}[\log(m_\phi)] - \mathbb{E}_p[\log(m_\phi)] \quad (2)$$

Looking at (2) we see that we need to calculate terms in the form of:

$$\mathbb{E}_g[\log(f)] \tag{3}$$

where $f$ and $g$ are probability density functions. Let $\mathbf{z}$ be a continuous random variable with finite expected value, finite variance, and probability density function $g$. For a set of $n$ samples $D_n = \mathbf{z}_1, ..., \mathbf{z}_n$, for a large $n$, the expected value $\mathbb{E}_g$ can be approximated by taking the sample mean. Hence we can rewrite (3) as:

$$\mathbb{E}_g[\log(f)] \approx \overline{\log(f)}^{(g)} \tag{4}$$

We use the bar symbol to denote the mean of a sample and we add $(g)$ to emphasize that the mean is taken from a sample with probability density function $g$. For each data point in $D_n$ we estimate $f$ using kernel density estimator (KDE):

$$\hat{f} = \frac{1}{lh} \sum_{i=1}^{l} K_h\left(\mathbf{z} - \mathbf{z}_i\right) \tag{5}$$

where $l$ is the number of observations, $h$ is the bandwidth parameter and $K$ is a scaled multivariate Gaussian kernel $K_h(\mathbf{u}) = (2\pi)^{-d/2} \exp\left\{-\frac{1}{2}\mathbf{u}^\top \mathbf{u}\right\}$. Choosing an estimator with a differentiable kernel is an essential property for constructing a differentiable loss function to be optimized later by a gradient descent algorithm.

Combining (4) and (5) we get:

$$\mathbb{E}_g[\log(f)] \approx \overline{\log(\hat{f})}^{(g)} \tag{6}$$

Note that $D_n$ is sampled with probability density function $g$ but for each data point, the value is estimated using KDE based on observations with underlying probability density function $f$

Inserting (6) into (2) we get:

$$L_{pdf} = \overline{\log(\hat{q}_\phi)}^{(q_\phi)} - \overline{\log(\hat{m}_\phi)}^{(q_\phi)} - \overline{\log(\hat{m}_\phi)}^{(p)} \tag{7}$$

In practice, during the training process, for each batch $L_n = \mathbf{z}_{q1}, ..., \mathbf{z}_{qn}$ of samples in the latent space, we generate a corresponding batch of $T_n = \mathbf{z}_{p1}, ..., \mathbf{z}_{pn}$ target points with a known probability density function. To calculate each term in (7), we implement a routine:

$$kd\_out = \text{kde\_fn}(obs, loc, bw) \tag{8}$$

Where $obs$ denotes the observations for density estimation for each data point, $loc$ denotes the locations of the data points, and $bw$ is a scalar that denotes the kernel bandwidth. The routine returns the average probability density function over all locations. To calculate the term $\overline{\log(\hat{q}_\phi)}^{(q_\phi)}$, both the locations and the observations are taken from $L_n$. To calculate the term $\overline{\log(\hat{m}_\phi)}^{(q_\phi)}$ the locations are taken from $L_n$, while $\frac{n}{2}$ observations are randomly sampled from $L_n$, and $\frac{n}{2}$ observations are randomly sampled from $T_n$. To calculate the term $\overline{\log(\hat{m}_\phi)}^{(p)}$ the locations are taken from $T_n$ while the observations are sampled from $L_n$ and $T_n$ as in the previous term. In all our experiments we select $bw$ using Scott's rule of thumb (Scott, 1992).

Intuitively, while KDE is known to be a poor estimator for multidimensional densities, we mitigate this issue by selecting a target function that is easier to estimate (e.g. Normal, GMM). Consequently, the initial estimation during the training process may be suboptimal, but it is still sufficient for the netwrok to converge. As training continues, the estimation gradually becomes more accurate.

While $L_{pdf}$ reshapes the statistical characteristics of the latent space, the system still has many degrees of freedom. For better control of the transformation of data points to the latent space, we add two additional terms. The first term, $L_{cl}$, enforces the clustering of specific data points within the

latent space. Equations (9) and (10) are two examples of such a term. The term in (9) penalizes data points that are located more than a distance $T$ from the center of their target cluster.

$$L_{cl} = \frac{1}{n} \sum_{i=1}^{n} \text{Relu} \left( \sum_{j=1}^{k} y_{i,j} \cdot \|\mathbf{z}_i - \mathbf{c}_j\|_2 - T \right) \tag{9}$$

In this equation, $k$ represents the number of predefined clusters in $\mathcal{Z}$, $C_j$ and $\mathbf{c}_j$ represents the j[th] cluster and it's corresponding centroid, and $T$ is a hyperparameter of the model that indicates the distance threshold. Additionally, $y_{i,j}$ is a boolean function that takes a value of 1 when data point $i$ belongs to cluster $j$, and 0 otherwise.

The term in (10) is aimed to enforce a separating hyperplane $z_{i,j,0} = 0$ where the indexes indicate the data-point, cluster and dimension respectively in a scenario of binary classification.

$$L_{cl} = \frac{1}{n} \sum_{i=1}^{n} \text{Relu} \left( \sum_{j=1}^{2} g_{i,j} \cdot \mathbf{z}_i \right) \tag{10}$$

In this equation, $g_{i,j}$ is a function that takes a value of 1 when data point $i$ belongs to cluster 1, and -1 when it belongs to cluster 2.

The last term is a reconstruction term, which encourages adjacent data points in the input space to remain adjacent in the latent space.

$$L_{rec} = \frac{1}{n} \sum_{i=1}^{n} \|\mathbf{x}_i - \hat{\mathbf{x}}_i\|_2 \tag{11}$$

The total loss $L_{tot}$ is then expressed as:

$$L_{tot} = \alpha L_{pdf} + \beta L_{cl} + (1 - \alpha - \beta) L_{rec} \tag{12}$$

where $\alpha \geq 0$ and $\beta \geq 0$ are hyperparameters of the model. Examples for the interplay between the three terms are discussed in details in the experimental results section.

## 2.2 ARCHITECTURE

A DeepKDE basic block consists of three consecutive layers: A linear layer followed by batch normalization (Ioffe & Szegedy, 2015) and Parametric Rectified Linear Unit (PReLU) (He et al., 2015b). The DeepKDE net architecture is constructed as an autoencoder and can be formally expressed by:

$$\begin{aligned}
\text{Block}(\mathbf{x}) &= \text{PReLU}(\text{BatchNorm}(\text{Linear}(\mathbf{x}))) \\
\text{Latent}(\mathbf{x}) &= \text{Linear}(\text{Block}(...\text{Block}(\mathbf{x}))) \\
\text{DeepKDE}(\mathbf{x}) &= \text{Linear}(\text{Latent}(\mathbf{x}))
\end{aligned} \tag{13}$$

## 3 EXPERIMENTS

In all experimental setups, a paired input dataset and target dataset of equal size are employed, generated either from a Normal or GMM distribution, depending on the specific experiment. The network architectures of the DeepKDE models utilized in all experiments are outlined in Table 1. For all conducted experiments, the training of DeepKDE models was performed using the Adam optimizer (Kingma & Ba, 2015). The batch size was set to 10000, and the learning rate was initially set to 0.01 for the first 500 epochs and subsequently reduced to 0.001 for the remaining training iterations.

Table 1: DeepKDE architectures implemented in the experimental setups. The DeepKDE architecture utilized in the second setup remained consistent across all primary networks and datasets, with the only variation being the input size.

| | Synthetic | ResNet-18 \| ViT (CIFAR-10) \| ViT (FMNIST) |
|---|---|---|
| Input | $1 \times 2$ | $1 \times 512 \mid 1 \times 768 \mid 1 \times 128$ |
| Encoder | $\begin{bmatrix} \text{Block}(2, 8) \\ \text{Block}(8, 32) \\ \text{Block}(32, 8) \\ \text{Linear}(8, 2) \end{bmatrix}$ | $\begin{bmatrix} \text{Block}(512, 1024) \\ \text{Block}(1024, 512) \\ \text{Block}(512, 256) \\ \text{Block}(256, 128) \\ \text{Linear}(128, 10) \end{bmatrix}$ |
| Latent | $1 \times 2$ | $1 \times 10$ |
| Decoder | $\begin{bmatrix} \text{Block}(2, 8) \\ \text{Block}(8, 32) \\ \text{Block}(32, 8) \\ \text{Linear}(8, 2) \end{bmatrix}$ | $\begin{bmatrix} \text{Block}(10, 128) \\ \text{Block}(128, 256) \\ \text{Block}(256, 512) \\ \text{Block}(512, 1024) \\ \text{Linear}(1024, 512) \end{bmatrix}$ |
| Output | $1 \times 2$ | $1 \times 10$ |

## 3.1 SYNTHETIC DATA

In this experiment we explore the basic properties of DeepKDE models by employing different target probability density functions, loss weights and $L_{cl}$ terms. The input data utilized in this study is a two-dimensional synthetic dataset, generated using the $make\_moons$ dataset from the scikit-learn 1.4.2 Python library (Pedregosa et al., 2011), with a noise parameter set to 0.05. A total of 10K data points are used for training, 10K for validation and 10K for testing. Figure 2(a) illustrates the input data points, with colors assigned to facilitate tracking in the latent space. Specifically, data points belonging to class 1 and class 2 are colored using the "cool" and "autumn" color maps from the matplotlib library (Hunter, 2007), respectively.

Figure 2(b) depicts the latent space in a scenario where the target dataset is generated from a normal distribution, specifically $\mathbf{z}_p \sim N(\mathbf{z}|\boldsymbol{\mu}, I)$ where $\boldsymbol{\mu} = \begin{pmatrix} 0 \\ 0 \end{pmatrix}$ and $I$ is the identity matrix. The supervision term $L_{cl}$ follows equation (10), and the weights assigned to $L_{pdf}$, $L_{cl}$, and $L_{rec}$ are set to $\{0.8, 0.2, 0\}$ respectively. Figure 2(c) shows the latent space where the target data points are generated from a distribution that is a mixture of two Gaussians, specifically $\mathbf{z}_p \sim \frac{1}{2} \sum_{i=1}^{2} N(\mathbf{z}|6 \cdot \mathbf{u}_i, I)$ where $\mathbf{u}_i$ is a unit vector and $I$ is the identity matrix. The weights in this setup were set to $\{0.5, 0.2, 0.3\}$. Figures 2(d)-2(f) provide some intuition on the significance of each term of the loss function. The target functions in these experiments remain the same as in Figure 2(c) but with different loss weights. Figure 2(d) depicts a model trained with weights $\{0, 0.9, 1\}$ emphasizing the role of $L_{pdf}$ in shaping the latent space to match the probability of the target data points. Figure 2(e) depicts a model trained with weights $\{0.9, 0, 0.1\}$ emphasizing the role of $L_{cl}$ in clustering points from the same class together in the latent space. Additional complementary experiments showed that omitting this term destabilizes the training process, resulting in mode collapse where all data points are concentrated within a single Gaussian in the latent space. Figure 2(f) depicts a model that was trained using weights $\{0.8, 0.2, 0\}$, emphasizing the role of $L_{rec}$ in encouraging adjacent data points in the input space to remain adjacent in the latent space.

## 3.2 CIFAR-10 & FASHION MNIST

In this experiment we test the effectiveness of our solution using well-known benchmark datasets - CIFAR-10 (Krizhevsky et al.) and Fashion MNIST (Xiao et al., 2017). Since the DeepKDE model

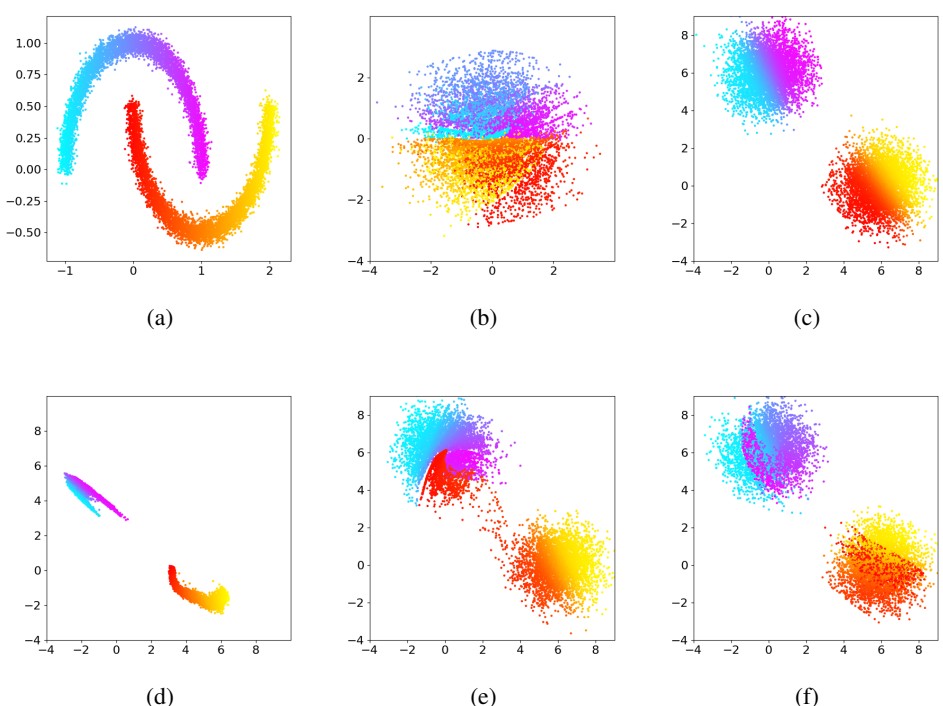

Figure 2: Analysis of DeepKDE models applied to two-dimensional input data. In 2(a), the input data points are displayed, with colors assigned to facilitate tracking in the latent space. 2(b) and 2(c) showcase scatter plots of DeepKDE latent spaces. 2(b) depicts a model trained to match a Normal distribution, while 2(c) presents a model trained to match a mixture of two Gaussians. Furthermore, 2(d)-2(f) depict the latent space of DeepKDE models that were trained using the same target function as in 2(c), but with varying loss weights. In 2(d), the assigned weights for $L_{pdf}$, $L_{cl}$, and $L_{rec}$ are $\{0, 0.9, 1\}$. In 2(e), the weights are $\{0.9, 0, 0.1\}$, and in 2(f), the weights are $\{0.8, 0.2, 0\}$.

should be independent of the source of its input, we evaluate its performance on inputs generated from two distinct primary classification models: ResNet-18 (He et al., 2015a), and a vision transformer (ViT) (Dosovitskiy et al., 2020). The output space's size of all models is $1 \times 10$ since both datastes have 10 different classes. The flow of this experiment starts with training the primary model and extracting features from a meaningful layer. These features are then provided as an input to a DeepKDE model with latent space of size $1 \times 10$. The target data points for the DeepKDE models in the experiment, are generated by mixture of 10 Gaussians located on the unit vectors:

$$\mathbf{z}_p \sim \frac{1}{10} \sum_{i=1}^{10} N(\mathbf{z}|6 \cdot \mathbf{u}_i, I) \tag{14}$$

where $\mathbf{u}_i$ is a unit vector and $I$ is the identity matrix. The loss weights in this experiment are set to $\{0.7, 0.1, 0.2\}$, and the classification is performed by assigning each data point to the cluster with the center positioned at the minimum Euclidean distance. The exact configurations of all primary models, training and feature extraction procedures can be found in Appendix A.

Figure 3 presents the accuracy scores of the DeepKDE models compared to the primary models for both data sets. The results show that DeepKDE outperforms the original models in terms of accuracy.

Figure 4 depicts scatter plots displaying two-dimensional projections of the ten-dimensional GMM shaped latent space. Figure 4(a) showcases the projections of the DeepKDE with ResNet-18 as the

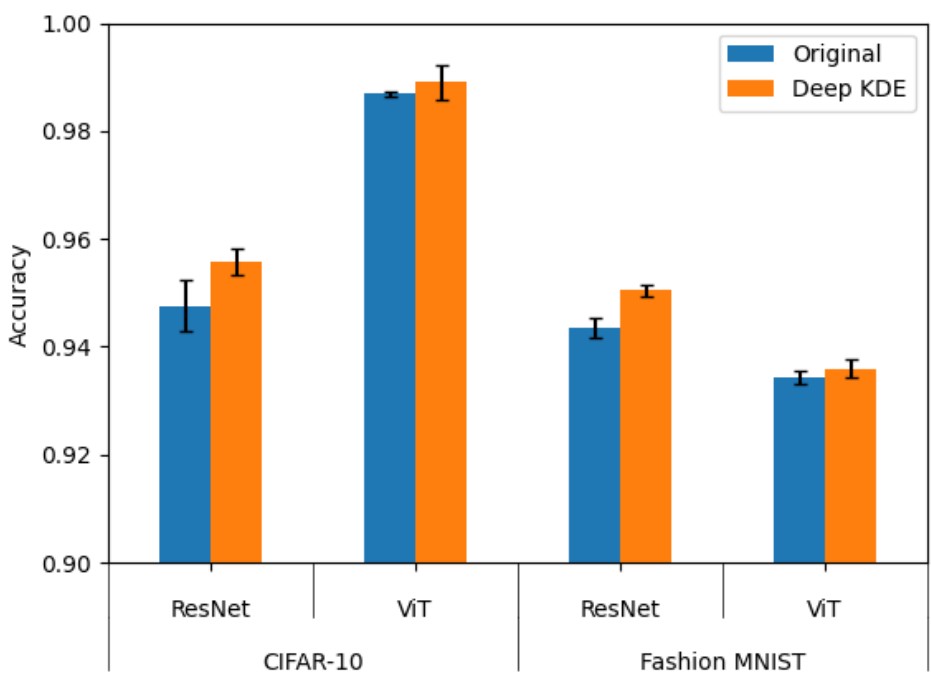

Figure 3: Accuracy scores of DeepKDE models compared to the original models for both data sets. The bars and error bars represent the mean scores and standard deviation over 10 repetitions. The Wilcoxon signed-rank test results for all experiments indicate statistically significant ($\alpha = 0.05$) differences in performance.

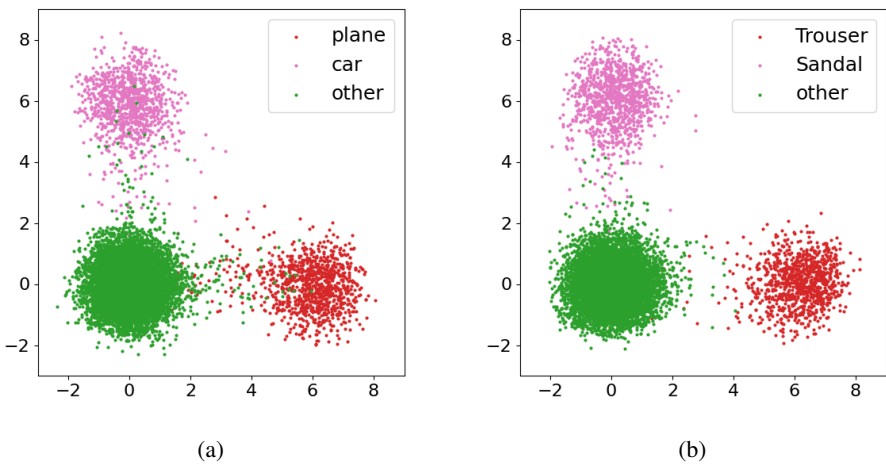

Figure 4: Scatter plots of two dimensional projections of the DeepKDE latent space for different models and datasets. Figure 4(a) showcases the projections on the axes corresponding to the plane and car classes of the CIFAR-10 dataset for the ResNet-18. Figure 4(b) showcases the projections on the axes corresponding to the Trouser and Sandal classes from the Fashion MNIST dataset

primary model, on the axes corresponding to the plane and car classes from the CIFAR-10 dataset. Figure 4(b) showcases the projections of the DeepKDE with ViT as a primary model, on the axes corresponding to the trousers and sandals classes from the Fashion MNIST dataset. In all figures, data points that have labels corresponding to one of the axes are colored in pink or red, while all other data points are colored in green. Specifically, there are 1K data points colored pink, 1K data points colored red, and 8K data points colored green in each figure.

Figure 5 illustrate further analysis of the results using our understanding of the latent space. The figure depicts the two-dimensional projection on the axes corresponding to cars and trucks classes from the CIFAR-10 dataset. We selected 6 data points from the center of each cluster (groups 1 and 5 in the figure), as well as 3 pairs of adjacent car-truck data points located in different areas (groups 2, 3, and 4). The images of the trucks are framed in cyan, while the images of the cars are framed in orange. By examining groups 1 and 5, we can observe that data points taken from the center of the cars or trucks distributions share similar characteristics and can be easily classified correctly. Group 2 demonstrates a case where a car is positioned within the trucks cluster. Upon observing the image of this car, we can see that it has a small trailer and its overall shape can be mistaken for a truck. Group 4 demonstrates the opposite scenario, where a truck is positioned within the cars cluster. Looking at the shape of the truck, it shares a closer resemblance with the neighboring car in comparison to the trucks. Group 3 comprises data points located far from the center of the clusters and provides examples of outliers.

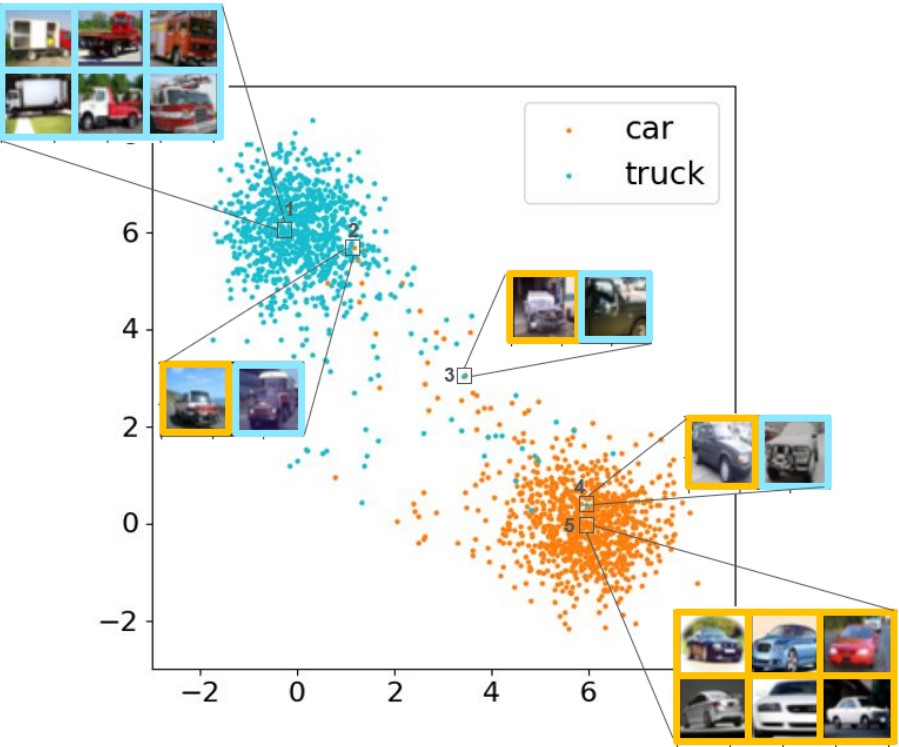

Figure 5: Cars and trucks classes from the CIFAR-10 dataset projected onto the corresponding axes in the DeepKDE latent space. Groups 1 and 6 consist of 6 data points sampled from the center of the trucks and cars clusters, respectively. Groups 2,3 and 4 consist of pairs of adjacent car-truck data points situated in different regions of the latent space. The images of trucks are highlighted with a cyan frame, while the images of cars are framed in orange.

## 4 CONCLUSIONS AND FUTURE WORK

In this paper we introduced the DeepKDE, a method for transforming the output space of classification models to match more desirable distributions, such as Normal and Gaussian Mixture Models.

We formulated the theoretical foundations of our method, explored its properties, and tested its effectiveness on ResNet-18 and ViT classification models trained on CIFAR-10 and Fashion MNIST datasets. Our results demonstrate that DeepKDE models succeed in transforming the data while enhancing the classification performance compared to the original models. Furthermore, the results can be further explained by using the knowledge about the underlying probability of the new space. Our aim in introducing this method is to establish a new application flow in downstream tasks such as anomaly detection and clustering.

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

## A  PRIMARY MODELS TRAINING AND FEATURE EXTRACTION

All models employed a consistent train-validation-test splitting method. Specifically, for the CIFAR-10 dataset, the official training data was split to 40K and 10K training and validation data respectively, while the 10K testing data remained consistent with the official dataset. Similarly, for the Fashion MNIST dataset, 50K and 10K images were divided from the official training data for training and validation, respectively, while the testing data remained consistent with the official dataset.

Feature extraction procedures in all models involved obtaining 300 instances of training and validation data features. This was done to enhance the DeepKDE training set. Additionally, one instance of testing data was extracted for DeepKDE evaluation.

## A.1 RESNET-18

**Train**

ResNet-18 was trained with the PyTorch 2.4 implementations of SGD optimizer with $lr = 0.01$, and CyclicLR scheduler (Smith, 2017) with $step\_size\_up = 10000$ and $step\_size\_down = 10000$. Batch size was set to 128 and number of epochs to 30. Augmentations were done using PyTorch's compose and transforms modules.

For the CIFAR-10 dataset, the following train, validation and test transformations were applied:

```
train_transform = transforms.Compose([
        transforms.Resize((224, 224)),
        transforms.RandomCrop(224, padding=4),
        transforms.RandomHorizontalFlip(),
        transforms.ToTensor(),
        transforms.Normalize((0.4914, 0.4822, 0.4465), (0.2023, 0.1994, 0.2010))
    ])

val_transform = transforms.Compose([
        transforms.Resize((224, 224)),
        transforms.RandomCrop(224, padding=4),
        transforms.RandomHorizontalFlip(),
        transforms.ToTensor(),
        transforms.Normalize((0.4914, 0.4822, 0.4465), (0.2023, 0.1994, 0.2010))
    ])

test_transform = transforms.Compose([
        transforms.Resize((224, 224)),
        transforms.ToTensor(),
        transforms.Normalize((0.4914, 0.4822, 0.4465), (0.2023, 0.1994, 0.2010))
    ])
```

For the Fashion MNIST dataset, the following train, validation and test transformations were applied:

```
train_transform = transforms.Compose([
        transforms.Resize((224, 224)),
        transforms.RandomCrop(224, padding=4),
        transforms.RandomHorizontalFlip(),
        transforms.ToTensor(),
        transforms.Normalize((0.5,), (0.5,)),
    ])

val_transform = transforms.Compose([
        transforms.Resize((224, 224)),
        transforms.RandomCrop(224, padding=4),
        transforms.RandomHorizontalFlip(),
        transforms.ToTensor(),
        transforms.Normalize((0.5,), (0.5,)),
    ])

test_transform = transforms.Compose([
        transforms.Resize((224, 224)),
        transforms.ToTensor(),
```

```
                    transforms.Normalize((0.5,),  (0.5,)),
            ])
```

**Feature extraction**

For ResNet-18, feature extraction was performed by removing the last layer before the fully connected layer, yielding a feature vector with a size of $1 \times 512$.

A.2    VISION TRANSFORMERS

**Train**

For CIFAR-10 dataset, we used the following implementation of the original ViT (Jeon, 2020). For Fashion MNIST dataset, we used a scaled-down version of the original ViT (schh, 2024). The training included fine-tuning of the pre-trained VIT-B16 network. Batch size was set to 64, number of epochs to 3000, warmup setup to 500, and learning rate to 3e-2. Augmentations were done using PyTorch's compose and transforms modules.

For the CIFAR-10 dataset, the following train, validation and test transformations were applied:

```
train_transform = transforms.Compose([
        transforms.RandomResizedCrop((224, 224), scale=(0.05, 1.0)),
        transforms.RandomCrop(224, padding=4),
        transforms.RandomHorizontalFlip(),
        transforms.ToTensor(),
        transforms.Normalize(mean=[0.5, 0.5, 0.5], std=[0.5, 0.5, 0.5]),
    ])
val_transform = transforms.Compose([
        transforms.RandomResizedCrop((224, 224), scale=(0.05, 1.0)),
        transforms.RandomCrop(224, padding=4),
        transforms.RandomHorizontalFlip(),
        transforms.ToTensor(),
        transforms.Normalize(mean=[0.5, 0.5, 0.5], std=[0.5, 0.5, 0.5]),
    ])
test_transform = transforms.Compose([
        transforms.Resize((224, 224)),
        transforms.ToTensor(),
        transforms.Normalize(mean=[0.5, 0.5, 0.5], std=[0.5, 0.5, 0.5]),
    ])
```

For the Fashion MNIST dataset, the following train, validation and test transformations were applied:

```
train_transform = transforms.Compose([
                transforms.Resize([224, 224]),
                transforms.RandomCrop(224, padding=2),
                transforms.RandomHorizontalFlip(),
                transforms.ToTensor(),
                transforms.Normalize([0.5], [0.5])])

val_transform = transforms.Compose([
                transforms.Resize([224, 224]),
                transforms.RandomCrop(224, padding=2),
                transforms.RandomHorizontalFlip(),
                transforms.ToTensor(),
                transforms.Normalize([0.5], [0.5])])

test_transform = transforms.Compose([
                transforms.Resize([224, 224]),
                transforms.ToTensor(),
                transforms.Normalize([0.5], [0.5])])
```

**Feature extraction**
For CIFAR-10 dataset, feature extraction was performed by extracting the first transformer vector, yielding a feature vector with a size of $1 \times 768$. Similarly, for the Fashion MNIST dataset, the same extraction process was applied, resulting in a feature vector of $1 \times 128$

