# OpenReview forum: "Reshaping Model Output Space Via Deep Kernel Density Estimation Networks"
_ICLR.cc/2025/Conference — ICLR 2025 Conference Withdrawn Submission_

### Official Review · Reviewer_dcBB · 2024-10-26

**Soundness:** 2
**Presentation:** 1
**Contribution:** 2
**Rating:** 3
**Confidence:** 3

**Summary:**

The authors introduce a novel method; Deep Kernel Density Estimation (DeepKDE) for transforming the feature spaces of arbitrary base models to known probability distributions, in order to obtain latent spaces with more desirable properties. DeepKDE can be attached to an arbitrary base model without the need for modifications to the original architecture. DeepKDE uses an auto-encoder-like architecture to map samples from the original feature space to a target probability density function by supervising in latent space with a divergence. The authors use a number of other loss terms to encourage samples of similar class to cluster together, and encourage samples that are close in the input space to be close in the transformed latent space. Experimental results are shown for synthetic data, as well as classification tasks on FashionMNIST and CIFAR10, where classification performance on the DeepKDE outputs consistently improves over the base model.

**Strengths:**

- The authors propose a (to my knowledge) novel and interesting method for transforming the latent spaces of arbitrary models to known probability density functions.
- The presented method shows some interesting experimental results; it seems to yield neatly clustered latent spaces.
- The paper contains illustrative figures that aid in understanding the paper.

**Weaknesses:**

- I have a hard time grasping the motivation for pursuing this method. In the introduction authors note that current models are "not designed to match any underlying probability distribution ... it would be advantageous to reshape the output space ... to align with a more convenient underlying probability function. This can enable downstream algorithms to benefit from the knowledge of the feature space distribution." The paper subsequently only investigates classification tasks, where I do not see a direct benefit from knowing the feature space distribution? Are classification models not optimized precisely for discriminative features? How is your method useful in this setting?
- To me, the method section is quite hard to follow. It seems that your proposed DeepKDE model consists mostly of pointwise operations (i.e. linear layers), how can this be applied to arbitrary deep learning models that have spatial (CNNs) or set-based (transformers) latents? Is a KDEBlock applied to each latent separately? Should information not be exchanged between spatial positions/set elements? I would appreciate some clarification on this.
- The experimental section is quite limited, with only results on synthetic data and two classification tasks on CIFAR10 and Fashion MNIST. Important details on the experimental setup are missing from the main body of the text, e.g. the ViT used by the authors presumably is a pretrained one, or it wouldn't obtain .99 test accuracy on CIFAR10? Are the results you show train test or validation accuracy? The addition of your DeepKDE to a baseline model also introduces new parameters and compute to the model, what would the baseline models performance be if you scale it up accordingly?
- The authors do not discuss computational complexity of their method. I think it is important to show this, as the method introduces an additional overhead to any baseline model, it is important to be able to gauge how significant this overhead is.

To me, the major limitation of this work is its lack of motivation. From the current manuscript and especially its experimental section, it is not clear to me why e.g. the latent space of a ViT needs to be mapped on to a GMM, what insights we might gain or what practical use-case this would have. I suggest the authors expand their introduction and experimental sections to include these considerations, and specifically show use-cases for their method that show real benefit from having a known latent space distribution.

**Questions:**

- In the introduction you discuss "output spaces" and "feature spaces", do these two concepts refer to the same thing? Is the point of DeepKDE to transform the latent space of a model or its output space? From the introduction this isn't quite clear to me.
- How do you choose what features from the base model you use for DeepKDE? Have you experimented with this?
- Could you provide details on the computational complexity that your model adds?
- How does your method work with set-latents or spatially arranged latent features (like in feature maps of CNNs)?

---

### Official Review · Reviewer_5epu · 2024-11-03

**Soundness:** 2
**Presentation:** 1
**Contribution:** 1
**Rating:** 1
**Confidence:** 4

**Summary:**

The paper proposes deepKDE to transform the output space of a deep network (extracted at some layer, or the last) to match some a priori distribution such as Normal and Gaussian Mixture Models. Some experiments are brought forward showing that the deepKDE loss does impact the configuration of the latent distribution.

**Strengths:**

The only strength of the submission is in the core motivation of finding better representation of Deep Networks that would match strong prior practitioners would have on their task. That premise albeit quite general is interesting and relevant in today's AI.

**Weaknesses:**

There are several weakness which led to my score.

**Experiments**

- no detail is provided to reproduce any of the experiments. Whether it is about the training of the models, the evaluation, the data processing, the data augmentations and so on and so forth. In short, nothing can be assessed from the experiments.

- the scope of the experiments is overly simple. CIFAR10 and FashionMNIST are overly simple datasets that are known to easily produce strong clustered representations with pretty much any type of training (starting from auto encoders). Anything that the authors claim should be tests at least on Imagenet100 if not Imagenet.

- complete absence of comparisons with prior work

**Contributions**

- the paper does not have any technical contribution. Other methods such as ``Deep Unsupervised Clustering with Gaussian Mixture Variational Autoencoders'' already propose to structure the latent space of a trained probabilistic model to exhibit clustering. That model can be used in pixel space or in latent space of pretrained backbones already. And this is only one among numerous recent papers trying to do just that (even if not always in a probabilistic setting)

- the paper completely fails to discuss prior work and existing methods that deal with similar ideas and to compare against them

**Questions:**

Based on the above concerns, I invite the authors to address as much of my concerns as possible although I strongly believe that at this point, the amount of work required would produce a new submissions altogether going out of the scope of a rebuttal's revision.

---

### Official Review · Reviewer_xDRj · 2024-11-04

**Soundness:** 3
**Presentation:** 2
**Contribution:** 2
**Rating:** 3
**Confidence:** 3

**Summary:**

In this paper, the authors propose an approach to reshape the intermediate feature space of a classification model to match a specified distribution. The paper proposes a new approach, Deep KDE networks, to achieve output reshaping. Deep KDE networks are inspired by an autoencoder architecture that uses the features from an intermediate layer of a pre-trained classifier to get representation space embeddings that match normal or GMM distributions. The authors use a combination of JS Divergence loss (using Kernel Density estimation to estimate the loss), a clustering loss, and a reconstruction loss to train the Deep KDE networks. Finally, the authors provide experimental results on a synthetic dataset and CIFAR10 and Fashion MNIST datasets to show the practical effectiveness of the proposed method.

**Strengths:**

The method shows some promise in improving the accuracy of the classification models and produces good clustering performance.

**Weaknesses:**

The motivation of the paper is not clear. The authors must clarify the setting under which they feel this might be advantageous. In a single-task setting, fitting a classifier should be easier than learning feature embeddings before fitting a classifier. In the multi-task setting, even if the feature distribution taken over all tasks follows a normal or GMM distribution, it is not necessary that the feature distribution of any particular downstream task would also follow a normal or GMM distribution.

The authors also do not compare their method against well-known representation learning methods to establish the relative advantage of having features follow a certain pre-specified distribution.

The experimental data also does not use SOTA classifiers as the SOTA accuracy on CIFAR 10, and FashionMNIST are higher than the reported values. Thus, it is unclear if the models considered have enough representational power to fit the data correctly. The performance improvements could also be explained by the increase in representational power given by the extra parameters in the Deep KDE model. It would be great if the authors could provide similar experiments using SOTA classifiers on CIFAR10 and MNIST datasets.

**Questions:**

Please refer to the Weaknesses section above.

---

### Note · Authors · 2024-11-27

I have read and agree with the venue's withdrawal policy on behalf of myself and my co-authors.